# Effect of Active-Edible Coating and Essential Oils on Lamb Patties Oxidation during Display

**DOI:** 10.3390/foods10020263

**Published:** 2021-01-27

**Authors:** Ana Carolina Pelaes Vital, Ana Guerrero, Pablo Guarnido, Izabella Cordeiro Severino, José Luis Olleta, Miguel Blasco, Ivanor Nunes do Prado, Filippo Maggi, María del Mar Campo

**Affiliations:** 1Department of Animal Husbandry and Food Science, Instituto Agroalimentario IA2, Universidad de Zaragoza-CITA, Miguel Servet 177, 50013 Zaragoza, Spain; ana_carolv@hotmail.com (A.C.P.V.); aguerre@unizar.es (A.G.); pabloguarnido@hotmail.com (P.G.); izaabellacordeiro@gmail.com (I.C.S.); olleta@unizar.es (J.L.O.); jblascos@unizar.es (M.B.); 2Department of Food Science, State University of Maringá, Av. Colombo, 5790, 87020–900 Maringá, Paraná, Brazil; 3Department of Animal Science, State University of Maringá, Av. Colombo, 5790, 87020–900 Maringá, Paraná, Brazil; inprado@uem.br; 4School of Pharmacy, University of Camerino, 62032 Camerino, Italy; filippo.maggi@unicam.it

**Keywords:** essential oil, antioxidant activity, oregano, thyme, alginate, ovine

## Abstract

The use of natural products to reduce the use of synthetic additives in meat products, reducing the oxidation and improving the shelf life is a current challenge. Meat quality from lamb patties during 10 days of display on modified atmosphere packaging (MAP) and active-edible coating were tested under six treatments: uncoated patties without coating (CON); patties with alginate coating (EC) and patties with coating and 0.1 or 0.05% of essential oils (EOs) from either thyme (TH 0.1; TH 0.05) or oregano (OR 0.1; OR 0.05). Display and treatment significantly modified (*P <* 0.001) all the studied meat quality variables (pH, color, water holding capacity, weight losses, thiobarbituric acid reactive substances (TBARS), antioxidant activity). Display produced discoloration and lipid oxidation, however, the samples with essential oils presented lower (*P <* 0.001) lipid oxidation than the CON or EC groups. Coated samples with or without EOs showed better color (lower lightness but higher redness and yellowness) and lower water losses (*P <* 0.001) than the CON. The addition of thyme EO caused a decrease (*P <* 0.001) in the consumer’s overall acceptability, whereas no statistical differences appeared between CON, EC and oregano EO addition. Thus, using EOs as natural antioxidants, especially those from oregano at low dosages (0.05%), could be considered a viable strategy to enhance the shelf life and the product quality of lamb meat patties without damaging the sensory acceptability.

## 1. Introduction

Shelf life is a basic aspect to consider in meat products due to their perishability. Lipid oxidation and color changes (discoloration) are between the main factors that decrease food quality, especially in meat products [1,2]. In fresh meat, color is an important indicator of wholesomeness that determines purchase or leads to product rejection [3].

With the aim of extending the shelf life of processed food products, synthetic additives are frequently used in the food industry. However, consumers have raised concerns with regard to the health risk of using these kinds of synthetics products. The increase in demand for more natural products encourages researchers to investigate new ways to achieve the challenge of preserving or increasing the shelf life with more natural and acceptable methods [1,4,5]. 

Among these methods, modified atmosphere packaging (MAP) stands out for keeping a desirable color in meat products frequently used by the meat industry. It can be done using gas mixtures associated with the packaging, and the effect on meat can also be improved by the addition of essential oils (EOs), encompassing active compounds endowed with antioxidant and antimicrobial properties [6].

EOs are natural products obtained from a large variety of plants. Many of them exhibit good potent antioxidant and/or antimicrobial properties which have been known since antiquity. In the last decade, EOs have presented an increased interest as natural additives in the food industry [7], many of them being considered as GRAS (generally recognized as safe) and approved by the Food and Drug Administration [8]. In addition, the European Commission registered some EO constituents, such as carvacrol, eugenol, thymol carvone, p-cymene, cinnamaldehyde, citral, limonene, and menthol to be used as food flavorings without any danger to the consumers´ health [9].

Recent reviews [5] have compiled the possible use of EOs to substitute synthetic antioxidants, preventing the oxidation and increasing the shelf life of meat products. However, many factors, such as the type of EO, dosage, method of incorporation into the products, type of meat product, among others, must be considered, encouraging the development of studies concerned with the efficiency and applicability of EOs as natural, innocuous alternatives in food preservation.

Oregano and thyme are worldwide spices and frequently used for culinary purposes by diverse cultures [10,11]. Familiarization with a product which is considered as safe and familiar improves consumer acceptability and a positive attitude in the use of them as replacers of synthetic food additives [12,13]. The antioxidant capacity of EOs has been connected to the presence of diverse compounds as thymol, carvacrol, *p*-cymene and γ-terpinene, among others [10,14], most of them notably present in the oregano and thyme EO composition [15,16,17].

Different combinations of the addition of suitable antioxidant agents and an appropriate packaging technique can minimalize losses of quality in meat products [4]. Uses of active alginate edible coatings are presenting great results in terms of meat quality preservation [14] and sensory acceptability [11]. Sensory attributes are highly modified by the addition of EOs due to their well-known aromatics’ characteristics, therefore, it is necessary to evaluate the consumer acceptability of the ‘new alternative’ meat products and find the ideal concentration of EO considering its effect on consumer preferences [5]. The addition of EOs in the edible coatings or films can be an effective strategy to limit their strong flavor disturbance [18]. In addition, EOs or other natural plant extracts can be added in different biopolymer films from other polysaccharides to improve their properties [19]. 

Ovine is a traditional sector that has experimented several changes in recent decades and a constant search for new strategies to increase its consumption. Ovine meat consumption has decreased in countries with a traditional consumption such as Mediterranean areas and has a great potential to increase in others such as Latin America. Among these strategies, the presentation of new meat commercial cuts and alternative modes of consumption to consumers is essential.

Thus, the aim of the current research was to evaluate the effects of edible and active coatings with oregano and thyme EOs on the quality attributes and sensory acceptability of lamb patties during refrigerated display under modified atmosphere packaging.

## 2. Materials and Methods 

### 2.1. Materials 

Folin–Ciocalteu, sodium carbonate, gallic acid, 2, 2-azinobis-3-ethylbenzotiazoline-6-sulfonic acid (ABTS), potassium persulfate, 2, 2-diphenyl- 1-picrylhydrazyl (DDPH), thiobarbituric acid (TBA), trichloroacetic acid (TCA) and 1,1,3,3 tetramethoxypropane (TMP) were from Sigma-Aldrich. Calcium chloride 2 hydrated was from PanReac (131232, Barcelona, Spain), sodium alginate from Biochemical (A3249, Barcelona, Spain) and the EOs were from Pranarôm International (Ghislenghien, Belgium). All reagents were of Analytical Grade.

### 2.2. Animals and Preparation of Lamb Patties with Coating

Meat was obtained from 16 commercial hind limbs of lambs from Rasa Aragonesa breed, selected from the cooperative Grupo Pastores^®^ among their animals labeled as “Ternasco de Aragón” Protected Geographical Indication Quality Label, slaughtered 24 h prior to the lamb patties elaboration. The average weight of the hind limbs was 1.41kg ± 0.075 kg.

Hind limbs were transported on the day of preparation under refrigerated transport to the Meat Quality Laboratory of Zaragoza (Veterinary Faculty, Spain) where they were deboned and the edible portion (muscle and fat) separated and ground (Gesame^®^ Mod 9432). The edible portion from all animals was mixed and homogenized. Patties were molded by a hand cutter (1 cm thickness) to weigh 55 g each, and randomly distributed to the different treatments tested, with 2 replicates (experiment) of 3 samples per each day of analyses per treatment. 

Edible coating was prepared with sodium alginate 2% (*w/v*) following methodologies described by Vital et al. [14]. For the active coating, the EOs of thyme (*Thymus vulgaris QT linalool*) and oregano (*Origanum vulgare*) from Pranarôm International^®^ (Ghislenghien -Belgium) were used. They were added at 0.1 or 0.05% (*w/w*) in their correspondent alginate coating and mixed under magnetic stirring. Chemical composition of EOs was analyzed by an Agilent 6890N gas chromatograph coupled with a 5973N mass spectrometer (Agilent, Santa Clara, CA, USA) and an auto-sampler 7863 (Agilent, Wilmingotn, DE, USA). The oven temperature was programmed following the method of Benelli et al. [20]. Helium (99.99%) was used as the carrier gas with a flow rate of 1 mL/min. Injector and detector temperatures were set to 280 °C. Two µL of the essential oil solution in *n*-hexane (1:100) was injected with a split ratio of 1:50. Peaks were acquired in full scan (EI mode, 70 eV) in the range of 28–400 *m/z*. α-Pinene, camphene, β-pinene, 1-octen-3-ol, myrcene, α-phellandrene, α-terpinene, p-cymene, limonene, 1,8-cineole, γ-terpinene, terpinolene, linalool, borneol, terpinen-4-ol, α-terpineol, thymol, carvacrol, (*E*)-caryophyllene, α-humulene and caryophyllene oxide were identified by comparison to the authentic standards (Sigma-Aldrich, Milan, Italy). The remaining compounds were assigned by using the combination of retention indices (RIs), calculated using a mixture of linear C_8_-C_30_ alkanes (Supelco, Bellefonte, CA, USA) according to the van den Dool and Kratz formula [21] and the similarity of the mass spectra (MS), with respect to those of the ADAMS, FFNSC2 and NIST17 libraries. Semi-quantitative values of the components were obtained by the peak area without correction factors. The values were the mean of three replicates. Table 1 compiles essential oils composition.

Lamb patties were equally and randomly divided into six groups: patties uncoated—control (CON); patties with edible coating (EC); patties with edible coating with 0.1% thyme EO (TH 0.1), patties with edible coating with 0.05% thyme EO (TH 0.05), patties with edible coating with 0.1% oregano EO (OR 0.1) and patties with edible coating with 0.05% oregano EO (OR 0.05).

All samples were packaged in an individual polystyrene tray with a modified atmosphere (70% O_2_ and 30% CO_2_) displayed under refrigeration conditions (between 2 and 4 °C) under light exposure (1200 lux, 12 h/ day), simulating market conditions. Samples of CON, EC, TH 0.1, TH 0.05, OR 0.1 and OR 0.05 were removed at 1, 3, 7 and 10 days of display for quality analysis.

Prior to the patties’ elaboration, the ground meat was analyzed and characterized according to several meat quality characteristics. 

### 2.3. Proximate Composition

The proximate composition of the ground meat used to elaborate the burgers was analyzed according to the standardized ISO protocols [22,23,24,25] for moisture, protein, fat, and ashes. Samples were taken the sampling day after homogenizing the ground meat, vacuum packaged and frozen (−20 °C). Then, they were transported to the Ingeniería y Servicios Cárnicos S.L. where the analyses were performed.

### 2.4. Fatty Acid Composition

Fatty acid analyses were performed after fat extraction described on Bligh and Dyer [26]. The methyl ester preparation included KOH in methanol, with C23:0 as an internal standard. To identify the methyl esters, a gas chromatograph (HP 6890) with a capillary column (100 m ×0.25 mm × 0.20 mm; SP 2380); was used [27]. The carrier gas was nitrogen. Samples were measured in duplicate and the results were expressed as the percentage of total fatty acids.

### 2.5. pH and Weight Losses

pH was registered each day of analysis on the patties using a pH meter (pH 7 portable pHmeter Lab Process) equipped with a penetration pH electrode. 

Weight of each patty was registered on the sampling day (after edible coat arrangement) and each day of the analysis, allowing to calculate the water holding capacity as the exudative losses percentage, during display according to the following equation: initial weight (day 0) − weight (analyses day)/initial weight) × 100.

### 2.6. Total Phenolic Compounds and Antioxidant Activity

Total phenolic compounds (TPCs) and antioxidant activity were measured on the EOs (1:1000 *v/v* with pure methanol) after their extraction from the patties samples (1:1 *w/v* with methanol), or the edible coatings (1:3 *v/v* with methanol). With homogenization and centrifugation (15 min, 4000 rpm) extracts were obtained. In the patties, filtration with filter paper was also performed.

#### 2.6.1. Total Phenolic Compounds (TPCs)

The TPCs were measured as described by Singleton and Rossi [28] with modifications. The sample (125 µL) was mixed with 125 µL of Folin–Ciocalteu reagent and 2250 µL of sodium carbonate (28 g/L). The samples reacted during 30 min in the dark (25 °C) and then the absorbance was read at 725 nm using a spectrophotometer (Onda^®^, Model: UV-20, Giogio Bormac Srl, Carpi (MO), Italy). The results were expressed as mg gallic acid equivalent (GAE) g of sample. The standard curve of gallic acid concentrations ranged from 0 to 300 mg/L.

#### 2.6.2. DPPH Radical Scavenging Assay

Protocols modified from Li et al. [29] were used to determined DPPH activity. A methanolic solution (2850 μL) containing DPPH (60 μM) was mixed with the samples (150 μL) during 30 min. Five hundred and fifteen nanometers (515 nm) was the absorbance wavelength used to read samples. Antioxidant activity was calculated as
DPPH activity (%) = (1−(Abs t/Abs t=0)) × 100
where: A sample t = 0: absorbance at time zero of the sample; A sample t: absorbance at 30 min of the sample.

#### 2.6.3. ABTS Radical Scavenging Assay

The ABTS activity was evaluated according to Re et al. [30] with modifications. ABTS· was obtained through the reaction between 7 mM ABTS (5 mL) and 140 mM potassium persulfate (88 μL), 16 h was the time necessary to incubate. The ABTS radical was mixed with ethanol (absorbance of 0.70 ± 0.02). Seven hundred and thirty-four nanometers (734 nm) was the wavelength used to record the scavenging activity (%). Samples (40 μL) were mixed with ABTS·+ solution (1960 μL) and absorbance was noted after 6 min of reaction. The scavenging activity (%) was calculated as
ABTS activity (%) = (1−(Abs t/Abs t=0)) × 100
where: A sample t = 0: absorbance of the sample at time zero; A sample t: absorbance of the sample at 6 min.

### 2.7. Lipid Oxidation Analysis

Lipid oxidation assays were performed by thiobarbituric acid reactive substances (TBARS), according to Pfalzgraf et al. [31]. Absorbance was measured at 532 nm with ONDA UV−20 spectrophotometer. The malonaldehyde (MDA) content was measured and the results were expressed as mg MDA/kg of meat. Lipid oxidation assays were assessed at 1, 3, 7 and 10 days of display.

### 2.8. Color Measurement

Color was determined by the CIEL*a*b* system at 1, 3, 7 and 10 days of display, using a Minolta CM-2002 (Konica-Minolta Business Solutions S.A., Madrid, Spain) spectrophotometer with a 10° view angle and a D65 illuminant, obtaining lightness (L*), redness (a*) and yellowness (b*).

### 2.9. Consumer Acceptability

Consumer tests were performed in a private room adequately adapted for sensory analysis at the University of Zaragoza (Spain). Eighty consumers were selected randomly within quotas of gender (46.25% women and 53.75% men) and age (36.25%: 18–25 years; 15.00%: 26–40 years; 26.25%: 41–55 years; 22.50%: >56 years) according to the Spanish national profile.

Eight sessions with ten different consumers were carried out. Each consumer assessed 6 different samples of patties, (one from each treatment evaluated), after 7 days of display. The samples were identified with a three-digit code and they were served in a randomized design, to prevent carry-over and order effects [32].

For culinary preparation, a pre-heated grill (200 °C) (SAMMIC^®^, P80-2) was used and each patty was individually cooked. Patties were covered with aluminum foil and cooked until reaching an internal temperature of 75 °C. Each sample was cut in five portions, wrapped in aluminum foil, and kept at 50 °C. Consumers were requested to taste the samples and evaluate the acceptability of different attributes (flavor, tenderness and overall acceptability) using a hedonic scale with 9 points which range from 1 (dislike extremely) to 9 (like extremely) without neutral central point (neither like nor dislike). They were informed to rinse their mouth with water and eat an unsalted tasted bread before evaluating each sample, including the first one.

### 2.10. Statistical Analysis

Meat quality attributes were assessed via the analysis of variance using the general linear model (GLM) procedures with SPSS (version 23.0) (IBM SPSS Statistics, SPSS Inc., Chicago, IL, USA) for Windows. Treatments and display were considered as fixed effects and their interactions were also considered. For consumer acceptability, treatment was the only fixed effect evaluated and the consumer was considered as a random effect. Differences between the means were evaluated using the Tukey test (*P* ≤ 0.050).

## 3. Results and Discussion

### 3.1. Essential Oil Composition

The *T. vulgaris* and *O. compactum* EO chemical constituents were reported in (Table 1). 

Oxygenated monoterpenes (67.4%) and monoterpene hydrocarbons (26.5%) were dominant in the *T. vulgaris* EO composition, with the monoterpene alcohol linalool as the most abundant component (48.5%). Other components occurring at percentage >5% were: δ-3-carene (6.8%), linalyl acetate (6.0%), α-pinene (5.6%) and limonene (5.1%).

The chemical composition of the *O. compactum* essential oil was characterized by oxygenated monoterpenes, accounting for 86.2%, whereas monoterpene hydrocarbons represented a minor fraction (11.7%) of the oil. The content of sesquiterpene fraction was negligible (1.4%). Among oxygenated monoterpenes, the phenolics carvacrol (65.4%) and thymol (18.5%) represented the two predominant compounds of the EO, accounting together for 83.9% of the total composition. Among the monoterpene hydrocarbons, the most representative compounds were *p*-cymene (6.3%) and γ-terpinene (3.9%), the biosynthetic precursors of thymol and carvacrol [33]. The remaining constituents identified in the EO were all in percentages < 1.3%.

The chemical profile of *T. vulgaris* belonged to the linalool chemotype, which is quite rare in this species. Notably, this chemotype has already been noticed in French populations of *T. vulgaris* [34]. On the other hand, the thymol and carvacrol chemotypes are characteristics in this species [35]. The *O. compactum* EO composition detected by us was qualitatively consistent with those previously found by other authors [36], though little quantitative variability can be observed. This can be related to harvesting time, plant processing, as well as genetic and geographic factors [37].

Carvacrol and thymol are two characteristic phenolic monoterpenes in several medicinal and aromatic plants belonging to Lamiaceae and Apiaceae families including the genera *Thymus*, *Origanum*, *Satureja*, *Ocimum*, *Thymbra*, *Trachyspermum* and *Oliveria* [38,39,40]. These compounds are reported as the most active EO components against bacterial and fungal growth.

### 3.2. Proximate Composition and Fatty Acids Analysis of Meat Matrix

The proximate composition (g/100 g) of the original mixed meat matrix presented 69.95% moisture, 19.0% protein, 6.5% fat, 0.78% carbohydrates and 1.05% ashes, having an energetic value of 164 kcal/100 g. Current results are comparable to those from the same Protected Geographical Indication ‘Ternasco de Aragón’ lamb legs characterized by [41]. However, they differ in terms of fat percentage, which also produces slighter variations on the fatty acid (FA) profile of the commercial legs between studies. The fatty acid profile of meat matrix was: 43.85% of SFA (saturated fatty acids), 46.61% of MUFA (monounsaturated FA) and 5.84% of PUFA (polyunsaturated FA). *n*-6 PUFA accounted 4.74% and *n*-3 PUFA 1.02% of total fatty acids. These results are typical of animals mainly fed on concentrates and slaughtered before three months of age, which is how most lambs are reared in Spain [42].

### 3.3. pH and Weight Loss of Lamb Patties

The pH and weight loss of patties are presented in (Table 2 and Table 3), respectively. There was an interaction (*P* < 0.001) between the treatment and display for pH. Until the third day, no differences were observed between treatments (*P* > 0.05). However, after this, the pH of the CON and EC treatments presented higher values than the other treatments with active coatings (with EO). This increase might be linked to meat spoilage, which results in a switch from a glycolytic to an amino acid-degrading microbial metabolism [43].

The coating decreased weight losses in the lamb patties during display (*P* < 0.001), and an interaction between the treatments and display time was also observed (*P* < 0.001). Although all treatments showed an increase in weight loss, this was more pronounced in CON treatments during all days evaluated. The coating acted as a barrier [44], keeping the water in the system, and thus, little or no exudate was released. This has also been observed in beef [14]. The formation of a gelatinous layer from the edible coating around the meat adhered after cooking [14] might help to reduce water losses. Other coatings have proved also to be effective in reducing water losses, such as whey proteins and monoacylglycerols [45] or chitosan with unsaturated fatty acids [46].

### 3.4. Antioxidant Activity of Essential Oils and Lamb Patties

In this study, TPC and antioxidant activity (DPPH and ABTS radical scavenging) were measured. The TPC value was 236.07 mg GAE/g for oregano EO and 22.5 mg GAE/g for thyme EO. The EO had a DPPH radical scavenging of 25.49 and 6.88% for oregano and thyme, respectively. The ABTS scavenging ability for oregano was 63.90 and 6.62% for thyme. Thus, thyme EO had lower TPC antioxidant activity (*P* < 0.001) than oregano EO. This result was also observed in the respective coatings with patties. Patties with coating + EO had higher antioxidant activity (*P* < 0.001) than control samples and those coated with oregano presented the highest antioxidant activity in all assessments. Related to ABTS, OR 0.1 presented an average of 28.69%, OR 0.05 of 25.87%, TH 0.1 of 24.40% and TH 0.05 23.70%. For DPPH, the values were 12.85% for OR 0.1, 12.28% for OR 0.05, 11.95% for TH 0.1 and 11.92% for TH 0.05. CON and EC had similar antioxidant activity (21.24% and 21.85% for DPPH, respectively; 11.46 and 11.49% for ABTS). Natural products with notable antioxidant activity have a good potential to be applied in meat industry, due to the content in active compounds that can reduce food deterioration during storage [47,48].

### 3.5. Lipid Oxidation of Lamb Patties

This assay measured the secondary products of oxidation, specially related to rancidity. The effect of coatings on the lipid oxidation of patties was evaluated throughout the display. The inclusion of EO in the coating and display time significantly influenced the TBARS values, and an interaction between these variables was observed (*P* < 0.001) (Table 2 and Table 4). Patties with coating containing EOs showed a lower value for oxidation, and coating + oregano EO was more efficient in reducing the lipid oxidation than coating with thyme EO. No difference was observed in relation to the EO concentration (0.05 or 0.1%). This difference between oregano and thyme EO might be associated to differences in their composition [49]. In oregano EO, 83.89% of the compounds were constituted by carvacrol and thymol (phenols), both of which have strong antioxidant activity while the essential oil of thyme has as a main compound the monoterpene linalool (54.53%), a major component with lower antioxidant activity than phenols. In this study, it was also possible to observe that the modified atmosphere (CON) was more effective to delay the lipid oxidation when compared with EC (modified atmosphere + coating) without EO. Oxidation is among the main factors in the deterioration of foods, leading to the rejection by the consumer and a decrease in quality. Thus, an edible coating with a natural antioxidant can enhance the shelf-life of meat products, through the prevention of lipid oxidation.

### 3.6. Color of Lamb Patties

The color can influence the consumer purchasing decisions [11], being one of the main factors at the time of purchase. In this way, it is important to verify the influence of coating on the color of hamburgers L*, a* and b* values are presented in (Table 2 and Table 5). L* values (lightness) decreased until day 3 for all treatments and then remained stable. CON showed higher L*, and this behavior may be associated with the highly oxidizing conditions, compared to samples with EO and coating, such as conformational changes in protein. The oxidative process may result in the rupture of peptides, protein–protein interactions and modification in amino acid chains. These changes may alter protein structure and function, leading to modifications of food attributes as color, texture and flavor. In addition, related to EC and compared to control, the presence of exudates in the coated patties (effect of coating) darkens the color. This behavior was observed in each coated treatment. Vital et al. [14] also observed an increase in L* when the coating was compared with high oxygen concentration (control sample). Related to a* (redness), CON presented a significant decrease (*P* < 0.001) over the storage and the coating decreased (*P* < 0.05) the color losses (compared to CON). In addition, at day 10, the a* values of the coated hamburger was >10, indicating a bright red color. The meat pigment, without oxygen, is in the form of deoxyMb (purple-red color). With air (O_2_), the pigment oxygenates (MbO_2_), with a bright red color. The oxygenation process was slowed down by the coating, and the coated treatments reached the maximum a* value between 7 and 10 days. b* value of CON was significantly different (*P* < 0.001) from the coated hamburgers, decreasing during storage in CON, while in the coated hamburger this parameter was not altered. Additionally, the coated samples exhibited the highest b* values, associated with the yellowish color of the coating. Coated treatments did not present a significant difference related to b*. Coating with EO can reduce color deterioration over the display time, extending the shelf-life of the meat products and making it more attractive to consumers.

### 3.7. Consumer Test

EOs have strong aromatic compounds whose presence can determine the specific aroma of plants and the flavor of condiments, one of their main functions also being to develop desirable flavors and aroma. The effect of treatment on consumer acceptability is compiled in (Table 6). Statistical differences (*P* < 0.001) were reported by consumers in terms of flavor and overall acceptability. No differences (*P* > 0.05) were pointed out with respect to tenderness acceptability, which confirms, as in other food products covered with alginate edible coating (beef steaks or fish fillets), that the presence of the coating did not decrease the texture acceptability [11,14,50]. 

Flavor acceptability of thyme EOs, especially at higher dosages (0.1%) was significantly lower than those reported in the CON group. The addition of oregano EOs decrease the acceptability score (*P* < 0.001). However, the differences were small related to CON and EC treatments.

Samples from TH 0.1 presented significantly lower overall acceptability that those from CON, EC or OR 0.1, treatments which did not statistically differ between them and obtained the highest overall acceptability scores.

Oregano EO had better acceptability than thyme for lamb patties. Higher dosages of oregano (0.1 vs. 0.05%) are slightly preferred by consumers. 

The choice of the EOs added and their concentration in a specific type of food is important because a small amount can cause sensory alterations, positive or negative, depending on both factors. The strong aroma of EOs can modify the food organoleptic properties [9]. Presumably the current results were more affected by the aromatic effect than by the antioxidant effect of EOs used, since the time of display (7 days) does not let strong undesirable off-flavors (rancidity) develop, which would be lower in EO samples with respect to CON or EC, as shown in the TBARS results (Table 4).

## 4. Conclusions

Edible coating (alginate-based) decreases the weight losses and discoloration of lamb patties. In addition, alginate coating is effective against lipid oxidation, an effect that is potentiated when EOs are added, which increase the antioxidant activity. Both concentrations of oregano EOs tested (0.1 and 0.05%) showed a higher antioxidant activity and lower lipid oxidation than those from the thyme EO.

Regarding consumer acceptability, patties with oregano were well accepted, as the CON and EC, while the patties with the highest concentration of thyme received lower notes. 

Thus, the combination of packaging (MAP) and alginate-based coatings with EO (considering the concentration and the type of EOs added) could be used in diverse meat products (such as lamb patties) in order to maintain or improve their shelf life, without adding undesirable sensorial characteristics to the product depending on the EO.

## Figures and Tables

**Table 1 foods-10-00263-t001:** Chemical composition of the essential oils from *Thymus vulgaris* and *Origanum compactum*.

No	Component ^a^	RI ^b^	RI Lit. ^c^	% *Thymus vulgaris* ^d^	% *Origanum compactum* ^d^	ID ^e^
1	tricyclene	915	921	Tr^f^		RI, MS
2	α-thujene	919	924	0.3 ± 0.1	0.1 ± 0.0	RI, MS
3	α-pinene	924	932	5.6 ± 1.2	0.2 ± 0.0	RI, MS, Std
4	α-fenchene	936	945	0.3 ± 0.0		RI, MS
5	camphene	938	946	0.2 ± 0.1	Tr ^f^	RI, MS, Std
6	sabinene	964	969	3.4 ± 0.7		RI, MS, Std
7	β-pinene	966	974	0.4 ± 0.1	Tr	RI, MS, Std
8	1-octen-3-ol	973	974	0.4 ± 0.1	0.1 ± 0.0	RI, MS, Std
9	3-octanone	985	979		0.1 ± 0.0	RI, MS
10	myrcene	987	988	0.9 ± 0.2	0.4 ± 0.1	RI, MS, Std
11	3-octanol	995	988	0.1 ± 0.0		RI, MS
12	α-phellandrene	1001	1002	Tr	Tr	RI, MS, Std
13	δ-3-carene	1006	1008	6.8 ± 1.3		RI, MS, Std
14	δ-terpinene	1007	1008		Tr	RI, MS
15	α-terpinene	1013	1014	0.7 ± 0.2	0.3 ± 0.0	RI, MS, Std
16	*p*-cymene	1020	1020	0.5 ± 0.1	6.3 ± 0.9	RI, MS, Std
17	limonene	1023	1024	5.1 ± 0.9	0.2 ± 0.0	RI, MS, Std
18	β-phellandrene	1024	1025		0.2 ± 0.0	RI, MS
19	1,8-cineole	1026	1025	0.1 ± 0.0	0.1 ± 0.0	RI, MS, Std
20	(*E*)-β-ocimene	1045	1044	Tr		RI, MS, Std
21	γ-terpinene	1054	1054	1.2 ± 0.2	3.9 ± 0.7	RI, MS, Std
22	*cis*-sabinene hydrate	1062	1065	0.9 ± 0.2		RI, MS
23	*cis*-linalool oxide	1069	1067	0.1 ± 0.0		RI, MS
24	*p*-mentha-2,4(8)-diene	1081	1085	0.1 ± 0.0		RI, MS
25	terpinolene	1083	1086	1.0 ± 0.3	Tr	RI, MS, Std
26	*trans*-linalool oxide	1085	1084	0.1 ± 0.0		RI, MS
27	*p*-cymenene	1086	1089		Tr	RI, MS
28	*trans*-sabinene hydrate	1094	1098	0.2 ± 0.1		RI, MS
29	linalool	1101	1095	48.5 ± 4.1	1.2 ± 0.3	RI, MS, Std
30	hotrienol	1104	1106	0.1 ± 0.0		RI, MS
31	*cis*-*p*-menth-2-en-1-ol	1117	1118	0.2 ± 0.0		RI, MS
32	*trans*-*p*-menth-2-en-1-ol	1135	1136	0.1 ± 0.0		RI, MS
33	camphor	1137	1141	0.3 ± 0.0		RI, MS, Std
34	*trans*-verbenol	1146	1140		Tr	RI, MS
35	borneol	1159	1165	0.3 ± 0.1	0.1 ± 0.0	RI, MS, Std
36	*p*-mentha-1,5-dien-8-ol	1163	1166	Tr		RI, MS
37	*cis*-linalool oxide	1166	1170	Tr		RI, MS
38	terpinen-4-ol	1171	1174	3.0 ± 0.6	0.5 ± 0.1	RI, MS, Std
39	*p*-cymen-8-ol	1182	1179	Tr	0.1 ± 0.0	RI, MS
40	α-terpineol	1185	1186	0.7 ± 0.2	0.2 ± 0.0	RI, MS, Std
41	*cis*-piperitol	1190	1196	0.1 ± 0.0		RI, MS
42	*cis*-dihydrocarvone	1193	1191	Tr	0.1 ± 0.0	RI, MS
43	*trans*-piperitol	1203	1207	Tr		RI, MS
44	*endo*-fenchyl acetate	1215	1218	0.1 ± 0.0		RI, MS
45	nerol	1226	1227	0.2 ± 0.0		RI, MS
46	citronellol	1229	1223	0.1 ± 0.0		RI, MS, Std
47	neral	1239	1235	Tr		RI, MS, Std
48	carvacrol, methyl ether	1241	1241	Tr	0.1 ± 0.0	RI, MS
49	piperitone	1249	1249	0.1 ± 0.0		RI, MS
50	linalyl acetate	1255	1254	6.0 ± 1.1		RI, MS
51	geranial	1269	1264	0.1 ± 0.0		RI, MS, Std
52	bornyl acetate	1280	1287	0.1 ± 0.0		RI, MS, Std
53	thymol	1293	1289	3.6 ± 0.7	18.5 ± 2.4	RI, MS, Std
54	carvacrol	1301	1298	0.4 ± 0.1	65.4 ± 3.9	RI, MS, Std
55	α-cubebene	1341	1345	0.1 ± 0.0		RI, MS
56	α-terpinyl acetate	1344	1346	1.3 ± 0.2		RI, MS
57	neryl acetate	1364	1359	Tr		RI, MS, Std
58	α-copaene	1367	1374	Tr		RI, MS
59	β-bourbonene	1373	1387	Tr		RI, MS
60	geranyl acetate	1383	1379	0.5 ± 0.1		RI, MS
61	(*E*)-caryophyllene	1405	1417	2.8 ± 0.5	0.5 ± 0.1	RI, MS, Std
62	α-humulene	1438	1452	0.1 ± 0.0	Tr	RI, MS, Std
63	*trans*-muurola-3,5-diene	1455	1451	0.4 ± 0.1		RI, MS
64	*cis*-muurola-4(14),5-diene	1466	1465	0.1 ± 0.0		RI, MS
65	*ar*-curcumene	1473	1479	0.1 ± 0.0		RI, MS
66	bicyclogermacrene	1480	1500	0.1 ± 0.0		RI, MS
67	α-zingiberene	1485	1493	Tr		RI, MS
68	δ-amorphene	1509	1511	0.1 ± 0.0	0.1 ± 0.0	RI, MS
69	hedycaryol	1536	1546	Tr		RI, MS
70	(*E*)-nerolidol	1555	1561	0.1 ± 0.0		RI, MS, Std
71	spathulenol	1560	1577	0.1 ± 0.0		RI, MS
72	caryophyllene oxide	1564	1583	0.6 ± 0.2	0.8 ± 0.2	RI, MS, Std
73	humulene epoxide	1591	1608		Tr	RI, MS
74	*epi*-α-bisabolol	1674	1683	Tr		RI, MS
75	eudesma-4(15),7-dien-1b-ol	1687	1687	Tr		RI, MS
76	isophyllocladene	1959	1967	Tr		RI, MS
77	manool oxide	1978	1987	Tr		RI, MS
78	nezukol	2124	2132	0.2 ± 0.0		RI, MS
	Total identified (%)			99.1	99.5	
	Grouped compounds (%)					
	Monoterpene hydrocarbons			26.5	11.7	
	Oxygenated monoterpenes			67.4	86.2	
	Sesquiterpene hydrocarbons			3.7	0.6	
	Oxygenated sesquiterpenes			0.8	0.8	
	Other compounds			0.7	0.2	

^a^ Elution order from an HP-5MS column. ^b^ Temperature-programmed linear retention index (RI) by the van den Dool and Kratz formula [21]. ^c^ Literature RI value obtained from ADAMS or NIST 17. ^d^ Value as the mean of three measurements ± SD. ^e^ Identification method: RI, correspondence of RI value with respect to those stored in NIST 17 or ADAMS libraries; MS, overlapping of the MS pattern with those recorded in NIST 17, WILEY 275, FFNSC2 and ADAMS libraries; Std, co-injection with analytical standard (Sigma-Aldrich, Milan, Italy). ^f^ Tr, traces, % < 0.05.

**Table 2 foods-10-00263-t002:** Meat characteristics of lamb patties with an edible coating and essential oils (EO) during the display period (mean ± standard error).

	Treatment (T)						Days of display (D)				T	D	T x D
	CON	EC	TH 0.05	TH 0.1	OR 0.05	OR 0.1	1	3	7	10			
pH	5.61 ± 0.04 ^a^	5.61 ± 0.03 ^a^	5.58 ± 0.02 ^b^	5.58 ± 0.02 ^b^	5.59 ± 0.02 ^b^	5.59 ± 0.02 ^b^	5.58 ± 0.01 ^b^	5.59 ± 0.01 ^b^	5.60 ± 0.01 ^a^	5.61 ± 0.02 ^a^	<0.001	<0.001	<0.001
Weight Losses †	7.23 ± 3.12 ^a^	3.00 ± 1.36 ^b^	3.17 ± 1.76 ^b^	3.03 ± 1.37 ^b^	3.37 ± 1.30 ^b^	3.23 ± 1.69 ^b^	1.80 ± 1.03 ^d^	2.76 ± 1.37 ^c^	5.05 ± 2.26 ^b^	5.75 ± 2.15 ^a^	<0.001	<0.001	<0.001
TBARS ††	0.27 ± 0.15 ^b^	0.34 ± 0.19 ^a^	0.19 ± 0.09 ^c^	0.18 ± 0.08 ^c^	0.10 ± 0.03 ^d^	0.09 ± 0.03 ^d^	0.08 ± 0.01 ^d^	0.13 ± 0.05 ^c^	0.28 ± 0.14 ^b^	0.31 ± 0.14 ^a^	<0.001	<0.001	<0.001
L*	48.62 ± 2.43 ^a^	43.00 ± 2.77 ^b^	43.09 ± 2.98 ^b^	43.49 ± 2.93 ^b^	42.94 ± 3.38 ^b^	42.83 ± 3.24 ^b^	47.88 ± 2.29 ^a^	42.99 ± 2.59 ^b^	42.80 ± 2.99 ^b^	42.33 ± 3.29 ^b^	<0.001	<0.001	0.321
a*	8.66 ± 1.29 ^b^	10.97 ± 1.28 ^a^	11.16 ± 1.54 ^a^	11.12 ± 1.33 ^a^	11.15 ± 1.02 ^a^	10.88 ± 1.56 ^a^	10.40 ± 0.98 ^a^	10.10 ± 1.13 ^a^	10.65 ± 1.66 ^a^	11.49 ± 2.09 ^b^	<0.001	0.001	<0.001
b*	13.99 ± 1.23 ^b^	19.29 ± 1.40 ^a^	19.31 ± 1.63 ^a^	18.99 ± 2.17 ^a^	18.43 ± 1.30 ^a^	18.791.16 ^a^	17.88 ± 1.49 ^b^	17.82 ± 1.98 ^b^	17.84 ± 2.82 ^b^	18.88 ± 2.91 ^a^	<0.001	<0.001	0.008

a, b, c, d: different lower letter means statistical differences in the same row within the treatment or within the days of display (*P* < 0.05). CON: patties uncoated; EC: patties with edible coating; TH 0.05: patties with edible coating with 0.05% thyme EO; TH 0.1: patties with edible coating with 0.1% thyme EO; OR 0.05: patties with edible coating with 0.05% oregano EO; OR 0.1: patties with edible coating with 0.1% oregano EO. †: % of weight losses; †† thiobarbituric acid reactive substances: mg malonaldehyde/kg of meat.

**Table 3 foods-10-00263-t003:** pH and weight losses evolution during the display period of the lamb patties with the edible coating and essential oils (mean ± standard error).

Days	CON	EC	TH 0.05	TH 0.1	OR 0.05	OR 0.1	*P* Value
pH							
1	5.59 ± 0.02B	5.57 ± 0.02C	5.58 ± 0.01B	5.58 ± 0.01B	5.58 ± 0.01B	5.58 ± 0.01B	0.210
3	5.59 ± 0.03B	5.59 ± 0.02B	5.58 ± 0.01B	5.58 ± 0.02B	5.59 ± 0.01AB	5.59 ± 0.01AB	0.149
7	5.64 ± 0.03Aa	5.64 ± 0.02Aa	5.57 ± 0.02Bb	5.58 ± 0.01Bb	5.59 ± 0.01ABb	5.60 ± 0.02ABb	<0.001
10	5.63 ± 0.02Aa	5.62 ± 0.01Aa	5.60 ± 0.01Ab	5.60 ± 0.01Ab	5.60 ± 0.01Ab	5.60 ± 0.01Ab	<0.001
*P* value	0.001	<0.001	<0.001	0.006	0.001	0.006	
Weight losses (%)							
1	3.54 ± 1.34Ba	1.59 ± 0.30Cb	1.34 ± 0.94Bb	1.39 ± 0.09Db	1.81 ± 0.18Bb	1.16 ± 0.28Bb	<0.001
3	5.28 ± 1.15Ba	1.91 ± 0.22Cb	1.90 ± 0.35Bb	2.29 ± 0.30Cb	2.93 ± 1.06Bb	2.24 ± 0.67Bb	<0.001
7	9.83 ± 0.89Aa	3.80 ± 0.40Bb	4.32 ± 0.92Ab	3.74 ± 0.35Bb	4.13 ± 0.84Ab	4.50 ± 0.13Ab	<0.001
10	10.27 ± 1.01Aa	4.72 ± 0.42Ab	5.11 ± 0.54Ab	4.71 ± 0.77Ab	4.63 ± 0.52Ab	5.03 ± 0.74Ab	<0.001
*P* value	<0.001	<0.001	<0.001	<0.001	<0.001	<0.001	

A, B, C: different upper letter means statistical differences between treatments within the display (*P* < 0.05). a, b, c: different lower letter means statistical differences between displays within the treatment (*P* < 0.05). CON: patties uncoated; EC: patties with edible coating; TH 0.05: patties with edible coating with 0.05% thyme EO; TH 0.1: patties with edible coating with 0.1% thyme EO; OR 0.05: patties with edible coating with 0.05% oregano EO; OR 0.1: patties with edible coating with 0.1% oregano EO.

**Table 4 foods-10-00263-t004:** Lipid oxidation evolution (mg malonaldehyde/kg meat) during the display period of the lamb patties with the edible coating and essential oils (mean ± standard error).

Days	CON	EC	TH 0.05	TH 0.1	OR 0.05	OR 0.1	*P* Value
1	0.08 ± 0.005Cbc	0.11 ± 0.021Ca	0.08 ± 0.010Cbc	0.08 ± 0.005Db	0.07 ± 0.006Cbc	0.06 ± 0.002Cc	<0.001
3	0.17 ± 0.020Bb	0.22 ± 0.016Ba	0.14 ± 0.021Bc	0.11 ± 0.011Cc	0.07 ± 0.005Cd	0.06 ± 0.004Cd	<0.001
7	0.42 ± 0.035Ab	0.50 ± 0.073Aa	0.26 ± 0.023Ac	0.25 ± 0.013Bc	0.12 ± 0.008Bd	0.11 ± 0.014Bd	<0.001
10	0.43 ± 0.061Ab	0.54 ± 0.036Aa	0.29 ± 0.036Ac	0.28 ± 0.021Ac	0.15 ± 0.015Ad	0.15 ± 0.012Ad	<0.001
*P* value	<0.001	<0.001	<0.001	<0.001	<0.001	<0.001	

A, B, C: different upper letter means statistical differences between the treatments within the display (*P* < 0.05). a, b, c: different lower letter means statistical differences between displays within the treatment (*P* < 0.05). CON: patties uncoated; EC: patties with edible coating; TH 0.05: patties with edible coating with 0.05% thyme EO; TH 0.1: patties with edible coating with 0.1% thyme EO; OR 0.05: patties with edible coating with 0.05% oregano EO; OR 0.1: patties with edible coating with 0.1% oregano EO.

**Table 5 foods-10-00263-t005:** Color evolution during the display period of the lamb patties with the edible coating and essential oils (mean ± standard error).

Days	CON	EC	TH 0.05	TH 0.1	OR 0.05	OR 0.1	*P* Value
*L**							
1	51.32 ± 2.71Aa	46.49 ± 1.29Ab	47.04 ± 1.38Ab	47.54 ± 1.26Ab	47.36 ± 1.64Ab	47.52 ± 1.86Ab	0.001
3	46.70 ± 2.04Ba	43.14 ± 1.77Bb	43.01 ± 1.86Bb	42.83 ± 1.84Bb	41.52 ± 2.59Bb	40.73 ± 0.99Bb	<0.001
7	48.17 ± 0.66Ba	40.93 ± 1.20Bb	41.78 ± 1.54Bb	42.55 ± 2.28Bb	40.74 ± 2.27Bb	42.63 ± 1.70Bb	<0.001
10	48.30 ± 1.34Ba	41.45 ± 2.54Bb	40.54 ± 2.42Bb	41.06 ± 1.06Bb	42.16 ± 2.37Bb	40.47 ± 1.63Bb	<0.001
*P* value	<0.003	<0.001	<0.001	<0.001	<0.001	0.001	
*a**							
1	10.13 ± 1.14A	10.72 ± 1.04AB	10.77 ± 1.06AB	10.48 ± 0.54B	10.81 ± 0.61AB	9.49 ± 1.00B	0.141
3	9.03 ± 0.86AB	9.67 ± 1.30B	10.05 ± 0.65B	10.67 ± 0.50B	10.33 ± 0.81B	10.83 ± 1.60AB	0.051
7	7.97 ± 0.45BCb	11.37 ± 0.66Aa	11.45 ± 1.94ABa	10.79 ± 1.43ABa	11.55 ± 0.86ABa	10.74 ± 1.10ABa	<0.001
10	7.52 ± 0.76Cb	12.12 ± 0.71Aa	12.37 ± 1.43Aa	12.55 ± 1.48Aa	11.92 ± 1.08Aa	12.45 ± 1.06Aa	<0.001
*P* value	<0.001	0.020	0.047	0.013	0.019	0.004	
*b**							
1	15.33 ± 0.69Ab	18.79 ± 0.98a	18.90 ± 1.23a	17.99 ± 1.17a	17.98 ± 0.77a	18.29 ± 0.65a	<0.001
3	15.61 ± 0.44Ab	18.41 ± 1.30a	18.73 ± 1.11a	18.63 ± 2.13a	18.15 ± 1.41a	18.40 ± 1.63a	<0.001
7	13.03 ± 0.84Bb	19.76 ± 1.66a	18.90 ± 2.42a	18.53 ± 2.85a	18.04 ± 1.22a	18.80 ± 1.19a	<0.001
10	12.99 ± 0.80Bb	20.20 ± 1.07a	20.69 ± 0.71a	20.82 ± 1.41a	19.57 ± 1.27a	19.68 ± 0.55a	<0.001
*P* value	<0.001	0.088	0.113	0.105	0.097	0.148	

A, B, C: different upper letter means statistical differences between treatments within display (*P* < 0.05). a, b, c: different lower letter means statistical differences between displays within treatment (*P* < 0.05). CON: patties uncoated; EC: patties with edible coating; TH 0.05: patties with edible coating with 0.05% thyme EO; TH 0.1: patties with edible coating with 0.1% thyme EO; OR 0.05: patties with edible coating with 0.05% oregano EO; OR 0.1: patties with edible coating with 0.1% oregano EO. The influence of coating on the color of hamburgers L*, a* and b*.

**Table 6 foods-10-00263-t006:** Consumer acceptability *(n* = 80) of lamb patties with edible coat and essential oils (mean ± standard error).

Acceptability ^1^	CON	EC	TH 0.05	TH 0.1	OR 0.05	OR 0.1	*P* Value
Flavor	6.87 ± 1.41 ^a^	6.75 ± 1.49 ^ab^	5.98 ± 1.79 ^bc^	5.45 ± 1.82 ^c^	6.21 ± 1.90 ^abc^	6.35 ± 1.86 ^ab^	<0.001
Tenderness	6.55 ± 1.56	6.95 ± 1.33	6.60 ± 1.69	6.46 ± 1.50	6.95 ± 1.47	6.80 ± 1.38	0.163
Overall	6.61 ± 1.41 ^a^	6.56 ± 1.57 ^a^	5.96 ± 1.73 ^ab^	5.57 ± 1.85 ^b^	6.26 ± 1.86 ^ab^	6.43 ± 1.67 ^a^	<0.001

^1^ Based on a 9-point scale from (1: ‘I dislike it extremely’, to 9: ‘I like it extremely’). a, b, c: different lower letter means statistical differences between the treatments (*P* < 0.05). CON: patties uncoated; EC: patties with edible coating; TH 0.05: patties with edible coating with 0.05% thyme EO; TH 0.1: patties with edible coating with 0.1% thyme EO; OR 0.05: patties with edible coating with 0.05% oregano EO; OR 0.1: patties with edible coating with 0.1% oregano EO.

## Data Availability

Data presented in this study are available on request from the corresponding author.

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
