# Peer review of "Effect of Active-Edible Coating and Essential Oils on Lamb Patties Oxidation during Display"

_foods, 2021, doi:10.3390/foods10020263_

Round 1
Reviewer 1 Report
This paper aimed to evaluate the use of edible coating using alginate, enriched with essential oils from thyme and oregano, known for their antioxidant properties. Moreover, a dose effect of essential oil is also considered.
The design of the experiment allowed to answer the research questioning. The results are clear and well presented. One can regret that for the consumer acceptability test, lamb patties were cooked (internal temperature of 75°C) and the authors did not report any appearance modification that could have interfered with acceptability assessment (color, coating desaggregated, taste?) due to heating.
It would have been interesting to include a consumer acceptability test on meat appearance since the Lab measurement highlighted a lightening of the meat coated, from the beginning (meat coated vs non coated) and a difference from day 10 to day 1 multiplied by 2 with coating (whatever essential oil is present or not)
It would have interesting to complete with microbiological measurement since the shelf life of a meat product include the sanitary aspect.
The introduction should cite Sulman & Poulson, 2013 Myoglobin Chemistry and Meat Color. Annual Review of Food Science and Technology, Vol. 4:79-99
https://doi.org/10.1146/annurev-food-030212-182623
The discussion about pH is not necessary since the increase of pH observed in very low, indeed significant, but the author should keep in mind that the error to the measure with such apparatus is 0.05 unit of pH. As far as I know pH was stable during display
Therefore one suggestion could be to focus the title of the paper on preventing oxidation during display rather than shelf life. Moreover, the low weight loss when meat coated is very interesting, maybe it could have been brought to light if a sensory test on juiciness was conducted. The tenderness of meat patties did not change whatever the conditions, which means that coating did not interfere with texture perception. May be other types of coating have other effect. The paper could be enriched with such discussion.
Author Response
Reviewer 1.
This paper aimed to evaluate the use of edible coating using alginate, enriched with essential oils from thyme and oregano, known for their antioxidant properties. Moreover, a dose effect of essential oil is also considered.
The design of the experiment allowed to answer the research questioning. The results are clear and well presented. One can regret that for the consumer acceptability test, lamb patties were cooked (internal temperature of 75°C) and the authors did not report any appearance modification that could have interfered with acceptability assessment (color, coating desaggregated, taste?) due to heating.
Thank you for the comment. We will consider appearance modification for future works. However, in the current project we did not have enough patties to assess visual appearance in independent patties besides those used for consumers’ tasting. In consumer trials the amount of sample is higher than in trials with trained panels. We were working fast enough for the patties to be tasted warm. A visual assessment of the same patties would have compromised the organoleptic properties in this case.
It would have been interesting to include a consumer acceptability test on meat appearance since the Lab measurement highlighted a lightening of the meat coated, from the beginning (meat coated vs non coated) and a difference from day 10 to day 1 multiplied by 2 with coating (whatever essential oil is present or not).
We agree with the comment, and as explained before, there was not enough meat to make more patties to be visually assessed independently from those patties used for tasting. For the tasting, patties were cut into portions, and covered in codified aluminium foil. The appearance was modified by the manipulation therefore we did not ask directly about the appearance. A visual evaluation in raw meat would be also interesting, but we lacked enough samples for this part as well.
It would have interesting to complete with microbiological measurement since the shelf life of a meat product include the sanitary aspect.
Again, this is an interesting point and we agree with the referee. For us, hygiene together with colour and flavour are the key points in meat shelf life. For this manuscript, we did not have more samples, but it is an aspect that we will develop in future works.
The introduction should cite Sulman & Poulson, 2013 Myoglobin Chemistry and Meat Color. Annual Review of Food Science and Technology, Vol. 4:79-99. https://doi.org/10.1146/annurev-food-030212-182623
We appreciate the suggestion of the article; it has been added to the text as reference number [3].
The discussion about pH is not necessary since the increase of pH observed in very low, indeed significant, but the author should keep in mind that the error to the measure with such apparatus is 0.05 unit of pH. As far as I know pH was stable during display
Therefore one suggestion could be to focus the title of the paper on preventing oxidation during display rather than shelf life.
We have modified the title accordingly: Effect of active-edible coating and essential oils on lamb patties oxidation during display.
Moreover, the low weight loss when meat coated is very interesting, maybe it could have been brought to light if a sensory test on juiciness was conducted. The tenderness of meat patties did not change whatever the conditions, which means that coating did not interfere with texture perception. May be other types of coating have other effect. The paper could be enriched with such discussion.
This is an interesting idea and has been added to the discussion, adding a couple of references that have worked in chitosan and whey protein coating. For this trial, we did not have more meat to do more patties, and we decided to perform consumer appraisal over panel assessment. Indeed, objective characteristics of juiciness would be better assessed with a trained panel.
Reviewer 2 Report
I have the following comments and suggestions:
- Lines 106-110: how meat was prepared, it should be explained more in detail. How it was packed?
- In the introduction part should be added the following reference since it is another possibility to make edible packaging: Jamróz, E., Kopel, P., Tkaczewska, J., Dordevic, D., Jancikova, S., Kulawik, P., ... & Adam, V. (2019). Nanocomposite Furcellaran Films—the Influence of Nanofillers on Functional Properties of Furcellaran Films and Effect on Linseed Oil Preservation. Polymers, 11(12), 2046.
- Table 1 should be in the Results and Discussion part.
- Lines 203-207: how TBARS was conducted, which device was used?
- Lines 291-302: where is the Table of mentioned results? Antioxidant activity for THY0.05.....
- Lines 308-309: Why did migration occur?
- Table 2: DAYS OF DISPLAY should be in shown as in the Table 3. It is not clear in the present written way.
- Table 3, 4, 5, statistical significance is presented with capital letters, why?
Author Response
I have the following comments and suggestions:
- Lines 106-110: how meat was prepared, it should be explained more in detail. How it was packed?
We are confused about the lines. We think that these sentences are referring to the current lines 99-104. We used fresh cuts, so they came to the laboratory without specific packaging under refrigeration in a box in the delivery truck. Afterwards in the lab were deboned, mix the edible portion and molded by a hand cutter as is compiled in lines 108-109.
Patties were briefly placed in their polystyrene tray with an oxygen film permeable until the moment of immersion into the alginate coating, and afterwards all samples were packaged with a modified atmosphere (70% O2 and 30% CO2) displayed under refrigeration conditions (between 2oC to 4oC) under light exposure (1200 lux, 12 h/ day), simulating market conditions as is compiled in the lines (139-141)
- In the introduction part should be added the following reference since it is another possibility to make edible packaging: Jamróz, E., Kopel, P., Tkaczewska, J., Dordevic, D., Jancikova, S., Kulawik, P., ... & Adam, V. (2019). Nanocomposite Furcellaran Films—the Influence of Nanofillers on Functional Properties of Furcellaran Films and Effect on Linseed Oil Preservation. Polymers, 11(12), 2046.
We appreciate the suggestion of the article; it has been added to the text as reference number [19]
- Table 1 should be in the Results and Discussion part.
It has been moved to Results and Discussion
- Lines 203-207: how TBARS was conducted, which device was used?
Absorbance was measured at 532 nm with ONDA UV-40 spectrophotometer, and this has been added into the text (Line 200-201). The technique is already published (Pfalzgraf et al. 1995, J. Agric. Food Chem. 43, 1339-1342. doi: 10.1021/jf00053a039), so we have not added more information to the text.
In any case, samples weighing 10 g ± 0.02 were homogenised with 20 mL of trichloroacetic acid (10%) using an Ultra-Turrax T25 (Janke & Kunkel, Staufen, Germany). Samples were centrifuged (Gyrozen 1248R, Daejeon, Korea) at 4000 rpm for 30 min and the supernatants filtered through qualitative paper (F1093 grade; Chmlab, Barcelona, Spain). Two millilitres of the filtrates were taken in duplicates and mixed with 2 mL of thiobarbituric acid, homogenised and incubated for 20 min in a water bath at 97 °C before absorbance was measured.
- Lines 291-302: where is the Table of mentioned results? Antioxidant activity for THY0.05.....
Due to the large number of tables of the manuscript, this section appears only as a detailed description of the results, not in a separate table. We have rewritten some part of the paragraph and added a supplementary sentence about thyme (line-305-310).
Lines 308-309: Why did migration occur?
We do not fully understand this comment. Colour is one of the most important characteristics when consumers are purchasing meat and meat products, especially when meat is packaged, either under vacuum or with modified atmosphere with high oxygen content. An attractive appearance is essential at this point, and the presence of oxymioglobin is very important in this attractiveness. This is what these sentences are referring to.
- Table 2: DAYS OF DISPLAY should be in shown as in the Table 3. It is not clear in the present written way.
Tables 3-5 are shown because there are significant interactions between days of display and treatment in most parameters (Table 2). Therefore, individual statistical analysis had to be performed per each treatment, and this is shown in Tables 3-5.
- Table 3, 4, 5, statistical significance is presented with capital letters, why?
We have written in capital letter differences due to the treatment within display, and in lower case those due to the display within treatment, for understanding better the differences. This has been added in the footnote for clarification.
Round 2
Reviewer 2 Report
The manuscript can be accepted.
Author Response
Authors would like to thank the suggestions for improving the manuscript.
Line 37: "On fresh meat color in..." On?
Done
Line 293: "proteins ad..." "and" instead of "ad"
Done
Line 407: "edible coatings... decreases"
Done
Lines 416-418. This sentence should be rewritten. It is not clear.
A modification has been done:
…could be used in diverse meat products (such as lamb patties) in order to maintain or improve their shelf life, without adding undesirable sensorial characteristics to the product depending on the EO